# A Study on Long–Close Distance Coordination Control Strategy for Litchi Picking

Hongjun Wang [1], Yiyan Lin [1], Xiujin Xu [1], Zhaoyi Chen [1], Zihao Wu [1] and Yunchao Tang [2,3,*]

[1] College of Engineering, South China Agricultural University, Guangzhou 510642, China; xtwhj@scau.edu.cn (H.W.); linyiyan@stu.scau.edu.cn (Y.L.); xiujin@stu.scau.edu.cn (X.X.); chenzhaoyi@stu.scau.edu.cn (Z.C.); wuzihao@stu.scau.edu.cn (Z.W.)

[2] College of Urban and Rural Construction, Zhongkai University of Agriculture and Engineering, Guangzhou 510006, China

[3] Foshan-Zhongke Innovation Research Institute of Intelligent Agriculture and Robotics, Foshan 528200, China

[*] Correspondence: ryan.twain@zhku.edu.cn

**Abstract:** For the automated robotic picking of bunch-type fruit, the strategy is to roughly determine the location of the bunches, plan the picking route from a remote location, and then locate the picking point precisely at a more appropriate, closer location. The latter can reduce the amount of information to be processed and obtain more precise and detailed features, thus improving the accuracy of the vision system. In this study, a long–close distance coordination control strategy for a litchi picking robot was proposed based on an Intel Realsense D435i camera combined with a point cloud map collected by the camera. The YOLOv5 object detection network and DBSCAN point cloud clustering method were used to determine the location of bunch fruits at a long distance to then deduce the sequence of picking. After reaching the close-distance position, the Mask RCNN instance segmentation method was used to segment the more distinctive bifurcate stems in the field of view. By processing segmentation masks, a dual reference model of "Point + Line" was proposed, which guided picking by the robotic arm. Compared with existing studies, this strategy took into account the advantages and disadvantages of depth cameras. By experimenting with the complete process, the density-clustering approach in long distance was able to classify different bunches at a closer distance, while a success rate of 88.46% was achieved during fruit-bearing branch locating. This was an exploratory work that provided a theoretical and technical reference for future research on fruit-picking robots.

**Keywords:** fruit-picking robot; litchi; computer vision; deep learning; segmentation

## 1. Introduction

Fruit-picking robots are a class of agricultural machines that combine the advantages of the accuracy, efficiency, and characteristics of diverse sensors. They primarily perform automatic operations for crops in natural environments [1–6]. Among them, the deployment of machine vision systems and corresponding recognition algorithms allows them to efficiently complete many harvesting operations. In current research, stereo vision techniques especially have been used by numerous researchers, while the flourishing of artificial intelligence and deep learning methods have provided better solutions for fruit recognition, thus allowing robots to adapt to orchard environments with complex backgrounds, uneven lighting, and low color contrast [7–11].

Further research on fruit picking has focused on how to build a compact, coordinated, and practical fruit-picking robot based on existing high-performance stereo vision systems and the algorithms suitable for that type of fruit object and its realistic environment [12]. There have been some representative cases in this regard. Wang et al. proposed

a litchi-picking robot based on a binocular stereo-vision-based system, which can separate and locate litchi in an unstructured realistic environment. The authors used an improved K-means clustering method to separate and a label template algorithm to locate fruit. Besides having good detection effects on nonoccluded and partially occluded litchi, their segmentation algorithm is robust to the influences of varying illumination [13]. Cao et al. solved the problem of obstacles, thus avoiding changing and unstructured environments for their litchi-harvesting robots. Their improved method, rapidly exploring random tree, was tested in both virtual and realistic environments, where two CCD cameras assisted in capturing scene information. The target gravity method and a genetic algorithm enhanced the speed and accuracy of the computation, thus successfully driving the manipulator to the target location without collision [14]. Wang et al. proposed a citrus fruit picking robot composed of a manipulator, binocular camera, personal computer, and tracked vehicle. They introduced a bite-mode end-effector with an improved harvesting postures prediction method and applied it to a fruit-picking robot. Through experimentations in both laboratory and natural environments, the harvesting results in the optimal posture had good performance [15]. Kalampokas et al. developed an autonomous grape harvesting robot that can reduce harvesting time by detecting grape stems in images. For this purpose, a regression convolutional neural network was deployed for finishing a stem segmentation task, which provides higher correct identification rates in highly changing environments [16]. Williams et al. introduced a new multi-arm kiwifruit harvesting robot working in pergola style orchards. The fruit picking robot consists of four picking robotic arms and corresponding end-effectors. With the assistance of deep neural networks as well as stereo matching methods, field implementation for detection and picking in commercial orchards was achieved [17]. In recent studies, increasing attention and effort for visual picking robots were shifted to working performance, including better environmental adaptability and stability of the harvesting process. Xiong et al. introduced a robot incorporating a CCD industrial camera with an LED illuminating source and formed a nocturnal image acquisition system for achieving the recognition of litchi and picking points in the nighttime environment. The YIQ color model was used and the fruits and stems were removed by an improved fuzzy clustering algorithm through the study of images collected in both day and night environments, with the final picking point determined mathematically [18]. Liang et al. also proposed a method for detecting litchi fruits and stems in the nighttime environment and achieved detection based on U-Net. In their work, after the processes to the bounding boxes of the YOLOv3, the regions of interest (ROI) of the stems were obtained. Through segmentation of the U-Net, the average precision in low-brightness can reach 89.30% [19]. Ye et al. introduced robot-arranged two-step collision-free motion planning based on the binocular stereo vision information. The spatial environment information was used to solve the robot's inverse kinematics problem using an improved adaptive weight particle swarm optimization algorithm to determine collision-free poses, while the improved Bi-RRT algorithm was used to achieve more accurate and faster path planning [20].

The above studies have successfully applied stereo vision in fruit-picking robots and considered the impact of various aspects on harvesting. However, most of them have used huge, cumbersome industrial cameras, which can easily damage fruit during harvesting and are expensive to replace. In addition, these systems require a lot of computing power and calculation time for spatial localization and stereo matching, with the algorithm for fruit stem recognition also being complex, and migrate poorly between different types of fruit. These problems have also led to difficulties in commercializing fruit-picking robots; four directions, such as simplifying the tasks and enhancing robots, have been proposed to guide researchers to make more attempts [21]. In summary, solving the above problems requires vision hardware that is more integrated, stable, and suitable to the unstructured environment of fruit picking. In addition, the development of more general algorithms based on this process is required [22].

The advent of depth cameras offers an economical solution for building three-dimensional (3D) optical coordinate measurement systems. Compact dimensions, low cost, and ease of secondary development yield such cameras, an increasingly popular hardware option for fruit-picking robots [23]. Liu et al. proposed a strategy for recognizing and locating citrus fruits in a close shot-based strategy on a Realsense F200 Camera (Intel Corp., Santa Clara, CA, USA). Using an intersection curve cut by the depth-sphere, six different fruit varieties were identified and the background removed, which might contribute to high real-time harvesting robots [24]. For harvesting in an irregular environment, Li et al. developed a reliable algorithm for detecting litchi fruit-bearing branches in large complex environments. Therein, Deeplabv3 was employed to distinguish the content of RGB images acquired by a Kinect V2 camera into three categories. The fruit-bearing branches belonging to the same cluster are identified by skeleton extraction, pruning operations, and spatial clustering. Finally, the positions of the branches are obtained using a 3D linear fitting [25]. Zhong et al. have used a YOLACT real-time instance segmentation network to locate fruit picking points based on the detection of litchi main fruit bearing branches (MFBB) and defined the robot picking posture by MFBB masking using skeleton extraction and least square fitting. The data obtained for the performance of test sets from different image sensors show that Kinect DK depth cameras can meet the needs of network training [26]. Yu et al. proposed a ripe litchi-recognition method using a Kinect V2 Red-Green-Blue Depth camera to capture multiple types of data. A random-forest binary classification model employing color and texture features was trained to recognize litchi fruit, whose dataset combined depth data and color data [27]. Fu et al. used a low-cost Kinect V2 camera to build an apple-picking background-separation vision system in a modern orchard to guide the robot in precise, collision-free harvesting. Using a deep learning network based on depth features to filter background objects, the authors demonstrated that the algorithm can improve fruit detection accuracy and is expected to be applicable for robotic harvesting on fruiting-wall apple orchards [28].

In fact, while consumer-grade depth cameras have made it easier to build vision systems for fruit-picking robots, there are still many hardware and depth-measurement technical limitations [29,30]. For example, point clouds obtained by depth cameras based on structured-light methods frequently include hole regions caused by missing depth information, which, along with lighting variations, have a large impact on image quality [31]. These drawbacks all lead to difficulties in acquiring high-quality, dense-point cloud data of fruit stems, making it difficult to research fruit-picking robots targeting fruit stems. Therefore, a fruit-picking system that takes the hardware and software conditions of the depth camera into account is urgently needed. Compared with traditional stereo vision, the depth camera's advantages in terms of shape and access to information can be exploited, while the picking point can be located within the allowed error range according to fruit characteristics.

The coordinated control of long-close distance has been proposed in other studies of picking strategies. Problems, such as complicated image information and insufficient representation of detailed features, have been solved. However, some solutions regard fruit bunches as a unit and cannot obtain information of the fruit number as well as their shape and some have low accuracy in acquiring the close-distance recognition position. Inspired by the work of other researchers, a litchi-picking robot was designed here based on the Intel Realsense D435i depth camera (Intel Corp., Santa Clara, CA, USA) and a long-close distance coordination control strategy proposed for bunches of fruit. The main contribution of this study included:

1. An eye-in-hand vision system was built for a consumer-grade depth camera and a long-close distance recognition strategy designed to meet actual positioning habits.
2. A density-based spatial point cloud clustering method was used to classify irregularly shaped fruit bunches, while a YOLOv5 convolutional neural network identified individual litchi.

3.  A MaskRCNN instance segmentation network was used to segment bunches and bifurcate stems in a scene and extract the fruit-bearing branch positions based on mask relationships.
4.  Depth reference points and fruit stalk positioning lines were introduced to reduce the impact of depth camera accuracy, thus enabling the position determination of the fruiting-bearing branches at a close distance.

The remainder of this study is organized as follows: In Section 2, the hand–eye coordination strategy and methods of recognition at different distances are briefly described. In Section 3, the experimental materials and methods are presented, followed by the results and corresponding subsequent analyses. Finally, the conclusions and future work are summarized in Section 4.

## 2. Materials and Methods

### 2.1. Overall Framework

An eye-in-hand vision system based on a D435i camera and 6-degree-of-freedom robotic arm (RUOBO Corp., Foshan, Guangdong, China) was constructed. The litchi picking robot system was comprised of two modules, namely an eye-in-hand system and hand–eye coordination control strategy. The strategy was divided into two processes, including fruit-cluster location and picking-point location (Figure 1).

The eye-in-hand system module was implemented based on the binocular visual stereo calibration method. The configuration of the image sensors and ease of using the official application programming interface (API) were taken into account, enabling determination of the correlation between the camera and robotic arm.

The hand–eye coordination control strategy module was divided into two steps. The first was to perform long-distance fruit cluster recognition, thus separating the clusters and obtaining close-distance recognition points. Second, taking into account the accuracy of the Realsense camera (Intel Corp., Santa Clara, CA, USA), the robot arm, being close to the bunch, obtained a close shot. The algorithm was then called to segment the litchi string and stem bifurcations, then predict the final picking point.

The position information obtained by locating the fruit bunch twice in the long- and close-distance enabled coordinated hand-eye control, and the method could be adapted to other types of bunch-fruit picking.

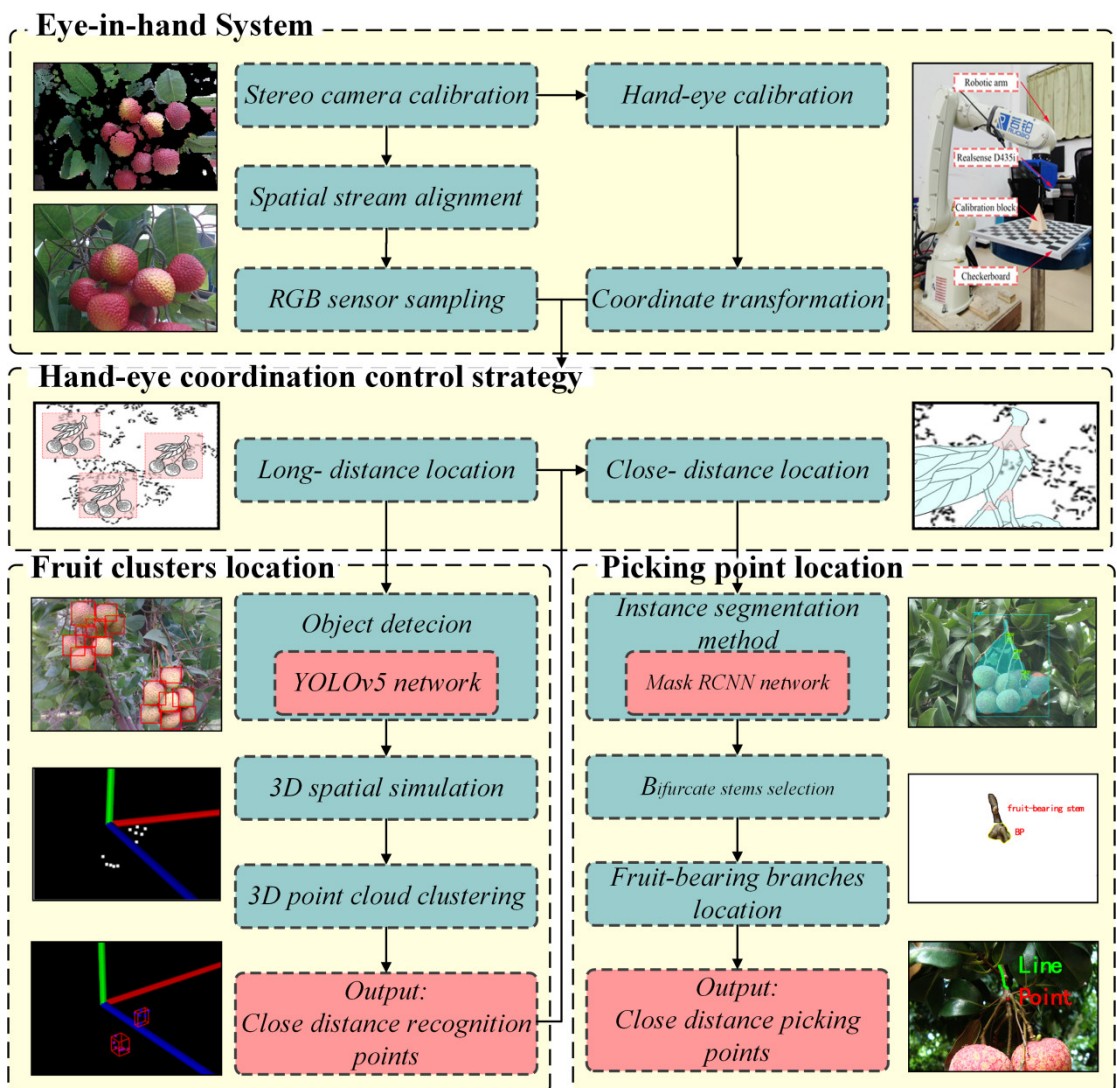

**Figure 1.** Litchi-picking robot system.

*2.2. Eye-in-Hand System*

Before sampling with the D435i depth camera, camera and hand–eye calibrations were required. Considering the sensor configuration of the depth camera (Figure 2), infrared radiation (IR) stereo cameras (Intel Corp., Santa Clara, CA, USA) were chosen for binocular calibration, implemented through the classic Zhang's method [32]. The hand–eye calibration was based on Chen's method, which obtained the coordinate transformation relationship of the left IR camera to the base of the robotic arm [12]. Due to the lack of color features in IR camera images and the API, the depth stream needed to be aligned to the color stream. From this, the coordinate transformation relationship was obtained between the two cameras to obtain a better-quality image for the construction of the deep-learning network dataset. Therefore, it was necessary to consider the coordinate transformation relationship between the RGB camera coordinate system and the left IR camera coordinate system during hand–eye calibration.

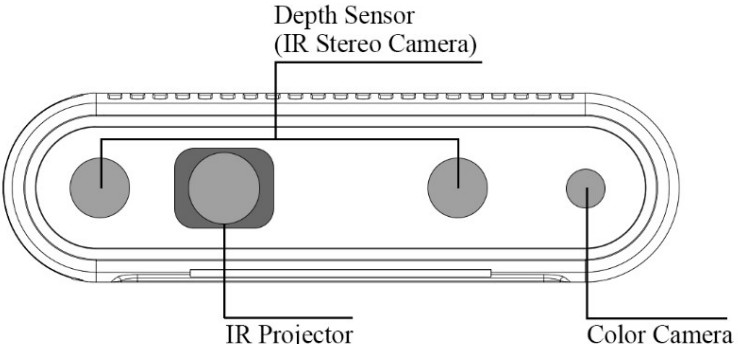

**Figure 2.** Sensor configuration of the Intel Realsense D435i depth camera.

A checkerboard was placed in front of the robotic arm as a reference for the calibration process. The relationship of each coordinate system is shown in Figure 3, with the two coordinate systems arranged on the depth camera. The coordinate system for each marker, described in turn, was {R} is the robotic arm coordinate system, {F} the flange coordinate system, {C} the left IR camera coordinate system, {RGB} the RGB camera coordinate system, {B} the checkerboard coordinate system, and {G} the end-effector coordinate system. The hand–eye calibration required constant adjustment of the arm's posture to allow the camera to sample from different positions, and it was seen that two of the coordinate system relationships were fixed for each movement. These were the relationships of {R} to {B} and {C} to {F}.

According to the basic principles of the coordinate transformation, the coordinate transformation chain was described as

$$_T^R T^{(i)} {}_C^F T {}_B^C T^{(i)} {}_R^B T = E,\tag{1}$$

where $E$ is the unit matrix. To obtain the coordinate transformation relationship between the tool (hand) and camera (eye), some transformations yielded

$$_C^F T = \left( {}_B^C T^{(i)} {}_R^B T {}_F^R T^{(i)} \right)^{-1},\tag{2}$$

where ${}_B^C T^{(i)}$ is the transformation matrix of {B} relative to {C}. As mentioned above, the robotic arm was adjusted to different $i$th postures from which a matrix was obtained by the least square method [33]. Then, ${}_R^B T$ is the transformation matrix of {R} relative to {B}, which was calibrated by the origin of {G} to approach the origin, and the $x$ and $y$-axes of {B}. Finally, the ${}_F^R T^{(i)}$ was recorded to obtain the transformation matrix of {F} relative to {R}, in every $i$th posture that could be calculated by forward kinematics.

Using the following calculations, the ${}_C^F T$ coordinate transformation relationship was obtained, expressed as

$$\widehat{{}_C^F T} = \frac{1}{N_p} \sum_{i=1}^{N_p} \left( {}_B^C T^{(i)} {}_R^B T {}_F^R T^{(i)} \right)^{-1},\tag{3}$$

where $N_p$ is the number of different poses. In the present method, $N_p$ was 15.

After calculating the hand–eye relationship, the coordinates of the target point in the left IR camera coordinate system ({C}) were obtained, and the position relative to the base coordinate system of the robotic arm ({R}) was calculated. The camera API provided a convenient function that allowed the selection of a point in the aligned image and the corresponding depth information obtained using the structured-light method, which meant that the aligned image was selected as the input for object detection.

However, the images acquired by the left IR camera were missing color information, and if the color stream was aligned to the depth stream, there would be many holes in the image. This would thus affect the training effect of the subsequent training of the convolutional neural network (CNN). Therefore, a transformation matrix from the color camera coordinate system ({RGB}) to the left IR camera coordinate system was introduced,

expressed as ${}_{RGB}^{C}T$. By checking the sensor configuration of the depth camera, the origin of the left IR camera coordinate system was found to be on the y-axis of the color camera coordinate system. The distance between these two origins of coordinates was 15 mm, such that it was possible to define ${}_{RGB}^{C}T$ as

$$
{}_{RGB}^{C}T = \begin{bmatrix} 1 & 0 & 0 & 0 \\ 0 & 1 & 0 & -15 \\ 0 & 0 & 1 & 0 \\ 0 & 0 & 0 & 1 \end{bmatrix},
\tag{4}
$$

Thus, the final hand–eye transformation relationship was

$$
T_{RGB}^{F} = {}_{C}^{\widehat{F}}T \, {}_{RGB}^{C}T,
\tag{5}
$$

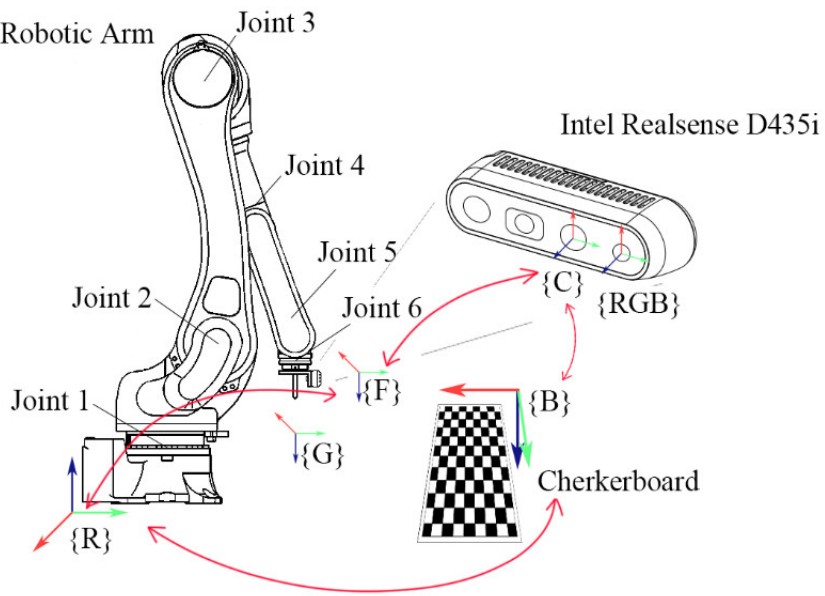

**Figure 3.** Coordinate transformation chain in the eye-in-hand system.

### 2.3. Hand-Eye Coordination Control Strategy

The D435i camera has a wider field of view than do D415 and D455 cameras and has a global shutter format that is more suitable for variable environments. This allows the D435i camera to capture more information at once about the environment and resist depth image blur. Comparing the parameters of these three cameras, the D435i camera has a smaller "min z" (distance from the depth camera to the captured object), which meant that the D435i camera could be used at a closer distance to the stems and thus had better detection performance. Thus, the D435i camera was chosen for building a robotic arm vision system based on the above comparison.

Servi et al. proposed a method for testing the effects of sampling distance to the imaging accuracy using different Realsense cameras, with results that illustrated that the farther the working distance is, the worse the results in image sampling. The structured-light measurement method also predetermines the working distance and condition of the target surface, which are two important influencing factors [34].

A two-stage hand–eye coordination control strategy was proposed based on such a situation (Figure 4).

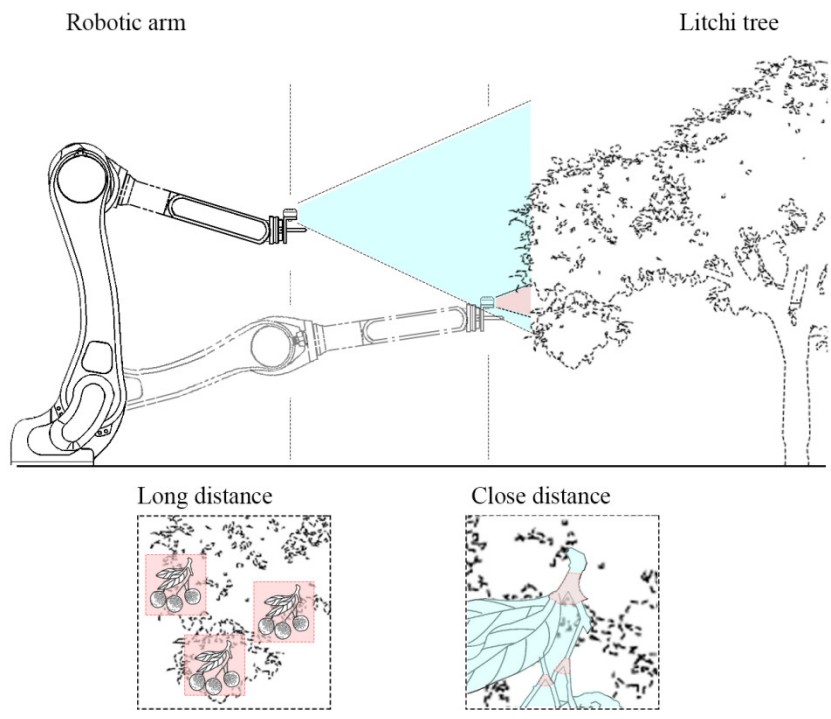

**Figure 4.** Two-stage hand–eye coordination control strategy.

Referring to the appropriate working-distance range of the D435i camera, the robotic arm first drove the camera at a long distance where it captured (identified) fruit bunches in a partial area of a tree in order to segment different fruit bunches and plan the picking route. After determining the close-distance recognition point, the robot arm was driven closer to the target bunch to identify the bifurcate stems and the whole litchi bunch. Compared with detecting a stem, bifurcate stems have more distinct features and thus are easier for study by neural networks.

Such a hand–eye coordination control strategy could be adapted to other bunch-fruit-picking operations, making full use to the high efficiency feature of the depth camera for achieving long-distance coarse positioning. The effects of the target scale can also be minimized, with the selection of segmentation bifurcate stems also improving the success rate, thus completing the operation within the error range of fruit picking.

### 2.4. Fruit Clusters Location

The first part of the hand–eye coordination control strategy was to perform long-distance location, in which the robotic arm was controlled far away from the bunches. The purpose at this stage was to locate and separate bunches in the field of view to determine the picking sequence as well as to guide the close-distance location.

The specific operation was to use the YOLOv5 convolutional neural network to detect individual litchi and obtain its bounding box. Then, we used the Realsense camera to obtain the depth information of the center of bounding box and simulate each litchi in 3D space. Next, density-based spatial clustering of applications with noise (DBSCAN) was used to separate the clusters, and a 3D minimum enclosing box was finally created for each cluster to determine its approximate location in space. The overall schematic is shown in Figure 5.

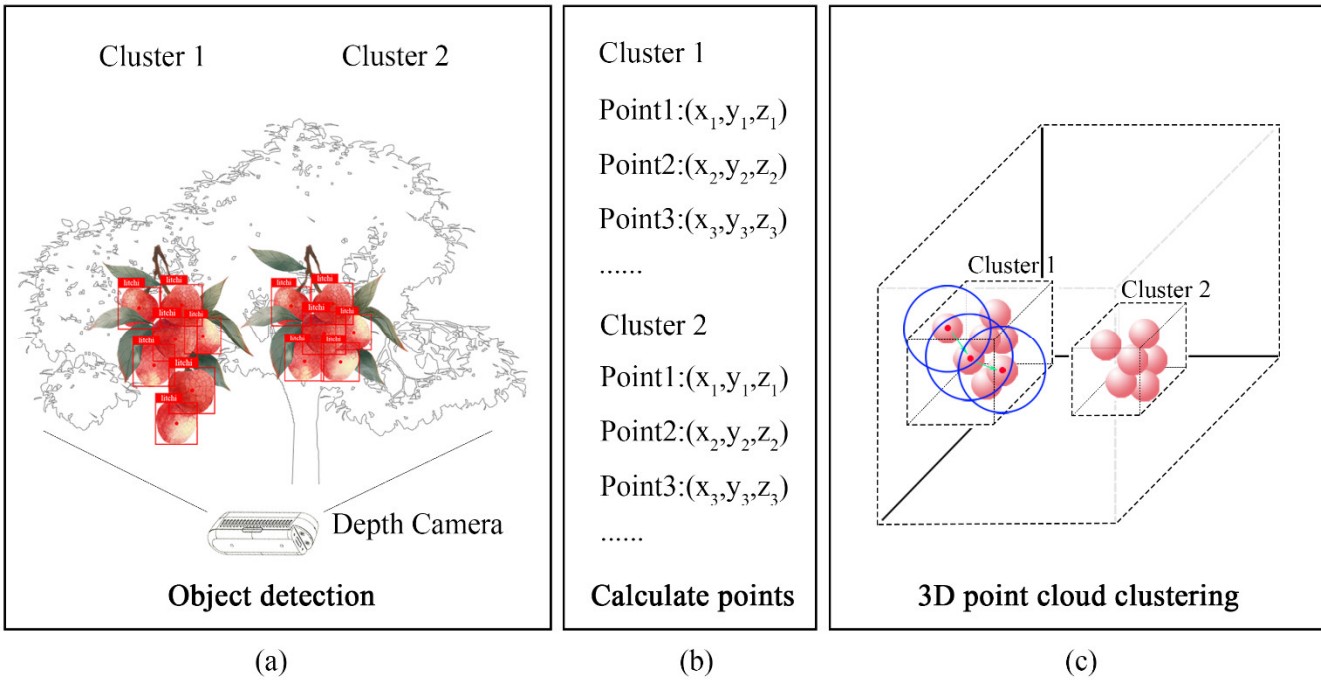

**Figure 5.** Long-distance location: (**a**) object detection; (**b**) calculated 3D points; (**c**) 3D point cloud clustering.

YOLO is a regression method based on deep learning, which has been developed through four versions from YOLOv1 to YOLOv4 [35,36]. Through continuous innovation and improvement, it has attained top performance. Here, the recent YOLOv5 network was chosen for the detection of litchi individuals by considering the aspects discussed below; the network architecture is shown in Figure 6.

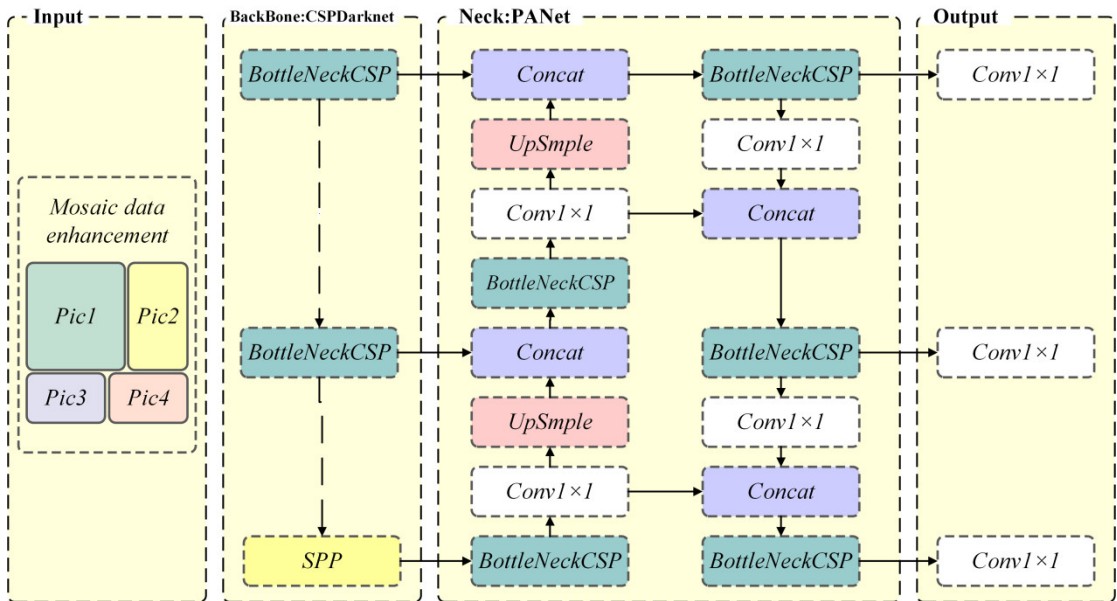

**Figure 6.** Structure of the YOLOv5 network.

Four features of YOLOv5 led to its selection as the target detection network. First, the input of YOLOv5 chose the same mosaic data enhancement method as YOLOv4. In long-distance location, the environment exists with different depth positions and different litchi sizes, which is quite advantageous for detecting litchi fruits with smaller scales using

this method. Second, YOLOv5 introduces two cross-state partial network structures, which allows the model to efficiently identify a large number of litchi individuals in the field of view. Immediately afterward, YOLOv5 introduces the structure of the feature pyramid network (FPN) and path aggregation as its neck, which enhances the flow of feature information in the network and is a significant improvement, especially for the learning of low-level features. Finally, the head of the YOLOv5 network outputs multi-scale feature maps, which, combined with the data enhancement method at the input, allows the network to handle litchis of different volumes.

After completing object detection, the camera was used to obtain the depth value of the center of each bounding box and the coordinates of each litchi at the camera coordinate system calculated based on the calibration parameters. For the subsequent spatial-point cloud clustering operation, the obtained coordinate information was input to the point cloud library (PCL) in the form of point cloud objects, and the attributes were set according to the average size of the real litchi fruit.

DBSCAN is a density-based spatial clustering algorithm that divides regions into clusters that reach the set parameters and finds clusters of arbitrary shapes in a spatial database with noise [37]. The algorithm has good results for dividing bunches of fruits with irregular shapes, as with litchi.

After determining the dataset to be clustered, two parameters were first initialized, namely *eps* and MinPts. Specifically, eps referred to the domain radius of the search, and MinPts the minimum number of points to determine a core point. It was also specified that all points of the dataset were classified into three types of points: core points, points that contain more than the number of MinPts within the range of eps; boundary points, points that have less than MinPts in the radius *eps* but fall in the neighborhood of the core points; and noise points, points that are neither core points nor boundary points.

First, any point in the dataset or the point with the largest neighborhood density was selected as the starting point, and if the point was not classified as a certain cluster or not a boundary point, it was determined whether it was a core point.

If the point was not a core point, it was classified as a boundary point or a noise point, and the starting point was reselected. If the point was a core point, it was classified into a new cluster, and all non-noise points in the neighborhood were grouped into this new cluster. After that, the points within this cluster that had not yet been classified were checked and classified in the same way until all points in the dataset were labeled.

In the reality of the harvesting process, different litchi bunches usually have touching, shading, and other spatial location relationships, which are not conducive to segmentation. At the same time, the shape of the bunches is random, such that the traditional clustering method based on Euclidean distance is prone to mis-segmentation. The DBSCAN method can adapt to various shapes of bunches and take into account the growth characteristics; also, the clustering results in a single direction reflect the complete fruit distribution characteristics. Moreover, this method is not affected by the shape of individual fruit and has good performance with other kinds of bunches of fruits.

After clustering, the 3D minimum enclosing boxes of different clusters was obtained, and the corresponding close-distance recognition points were calculated.

### 2.5. Picking Point Location

After the robotic arm drove the camera close to the close-distance recognition point, the next step was to identify the fruiting-bearing branches of that litchi bunch. Due to the small scale and lack of features, it was difficult to make a direct identification. Meanwhile, the Realsense camera was prone to returning wrong results when acquiring depth information of fine objects due to the method of structured light. All these reasons indicated that it was not feasible to directly acquire the spatial information of fruiting-bearing branches.

From the characteristics of the camera and the shape of litchi bunches, the instance segmentation network MaskRCNN was first used to segment the litchi bunches with picking stems in the field of view. Additionally, the bifurcate stems with obvious features and large scales were detected. After that, the bifurcate stems were filtered to determine the main bifurcate stems of the center cluster and to locate the fruit-bearing branch's mask. Then, the mask was extracted using Zhang's thinning algorithm and the points on the skeleton line obtained using the Realsense camera. Finally, a "Point + Line" spatial point cloud was formed with the center point of the main bifurcate stem and a spatial straight line was fitted to obtain the spatial position of the fruit-bearing branch. A schematic diagram of the above process is shown in Figure 7 [38].

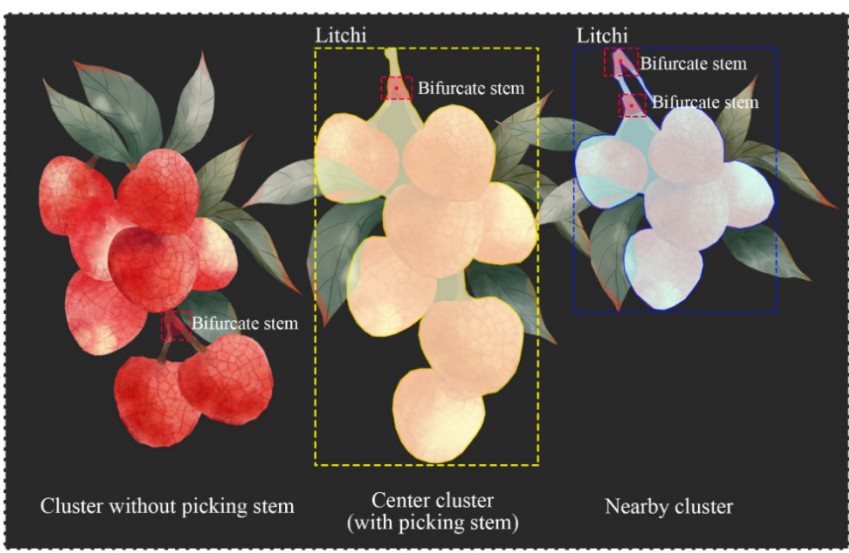

**Figure 7.** Picking point location.

MaskRCNN is an instance segmentation network, which adds a branch of prediction segmentation mask to FasterRCNN, replacing the ROI pooling layer with an ROI align layer and adding a parallel FCN layer (mask layer). The structure of the network is shown in Figure 8 [39,40].

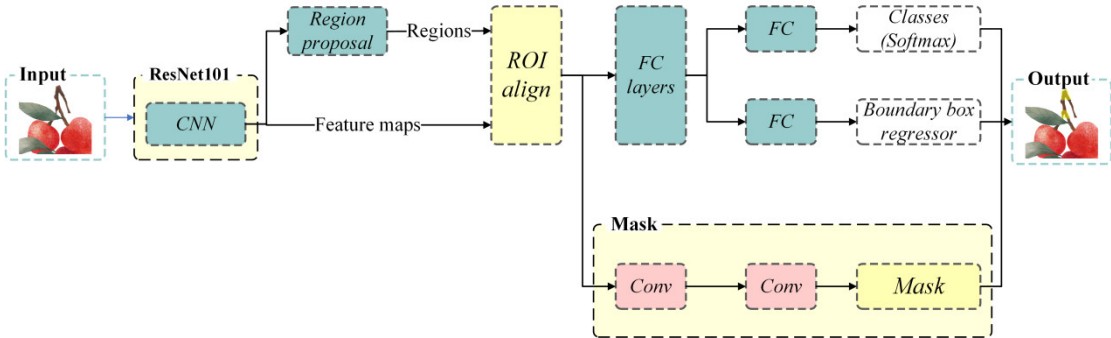

**Figure 8.** Structure of the MaskRCNN network.

MaskRCNN introduced the ROI align layer to obtain image values on the pixel points with floating point coordinates, thus improving the detection accuracy and facilitating segmentation of fine bifurcate stems. The backbone used a FPN network, which improved detection accuracy by inputting a single-scale image and finally obtaining a corresponding feature pyramid. These features make the MaskRCNN network work stably in the segmentation of litchi strings and bifurcate stems.

As long-distance location brought the camera close to a certain bunch of litchi, after segmenting the different litchi bunches and bifurcate stems in the scene, the bunch was selected whose center of mask was closest to the center of the image. All bifurcate stems whose center points were not within the bunch mask were then deleted, yielding the bifurcate stems belonging to that bunch. Then, the uppermost bifurcate stem was selected as the main bifurcate stem, and the part of the bunch mask above it was extracted as the fruit-bearing branch area.

After obtaining the fruit-bearing branch mask, the skeleton of the mask was extracted using Zhang's thinning algorithm, which was used to refine the fruit-bearing stem's mask and obtain a reference line. As the main bifurcate stem had a large scale, the depth value obtained using Realsense was more reliable such that the center point of the main bifurcation branch was used as a depth reference point. The resulting positioning line finally obtained the poses of the fruit-bearing branches to guide the final picking work. This method reduced the impact of outliers on the results, which was very helpful for reducing the impact caused by camera error sampling. The schematic is show as Figure 9.

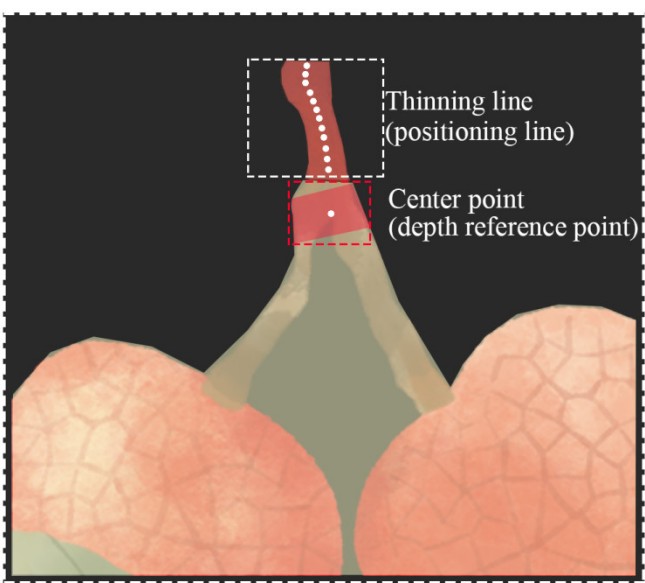

**Figure 9.** "Point + Line" location mode.

This method took into account the importance of the bifurcate stem of the fruit pedicel for position location of the fruit pedicel and the accuracy of the depth camera for obtaining depth. This used more reliable "points" to help fit more accurate spatial straight lines. Overall, such a method was feasible for most bunches of fruit.

## 3. Experiment

This section is divided into two parts: hand–eye coordinated control system evaluation experiments, and long-close distance localization and picking experiments. In the hand–eye coordinated control-system evaluation experiments, the hand–eye system using the D435i camera was mainly used for positioning accuracy and operating time experiments so as to verify the feasibility of the depth camera in litchi fruit picking and the performance of API. Experiments of long-close distance localization and picking were conducted to obtain the close-distance recognition point, along with the depth reference point and stalk positioning line of the fruit-bearing stem.

### 3.1. Hand–Eye Coordinated Control System Evaluation Experiments

Hand–eye calibration enables the camera to detect the target position in the image and then convert the pixel coordinates of the camera to the spatial coordinate system of

the robotic arm. Then, through the calibrated coordinate conversion matrix, the robotic arm is controlled to reach the specified position. However, the performance of the hand-eye system is affected by the repeated positioning errors of the robotic arm, camera recognition error, hand–eye calibration error, and tool calibration error such that these errors are complex and coupled [10].

In this experiment, two aspects of the hand–eye coordinated control system were evaluated. The positioning accuracy of the eye-in-hand system was determined by collecting the positioning errors of the corner points of a high-precision checkerboard. The performance of the depth camera was evaluated by recording the time consumption for obtaining the depth of the corners in the camera coordinate system. The experimental checkerboard had 88 corners, and the checkerboard placed horizontally in front of the robot arm at ~500 mm such that the *z*-axis of the robot arm coordinate system and checkerboard coordinate system were parallel to each other.

For each corner, the following coordinate transformation relations were used, expressed as

$$
{}^{R}P_{ij} = {}^{R}_{F}T_{i}{}^{F}_{C}T\,{}^{C}_{RGB}T\,{}^{RGB}P_{ij}, \tag{6}
$$

where ${}^{RGB}P_{ij}$ and ${}^{R}P_{ij}$ are the coordinates of the $j$th corner point at the $i$th shooting position relative to the depth camera color sensor coordinate system ({RGB}) and the robotic arm base coordinate system ({R}). Due to the use of the structured-light method for obtaining depth information, the depth camera had a faster corresponding speed in calculating ${}^{RGB}P$, compared with the conventional binocular vision system, using stereo matching. The ${}^{C}_{RGB}T$ is the conversion matrix obtained from the depth camera sensor distribution; ${}^{F}_{C}T$ the output of hand–eye calibration; ${}^{R}_{F}T_{i}$ is the conversion matrix of {F} relative to {R} at the $i$th pose of the robotic arm. For the $j$th corner on the checkerboard, the average of the 3D coordinates sampled for different robotic arm poses at the same sampling distance was considered the best estimate. The average Euclidean distance between each sample and the optimal estimate was considered the error at that distance, which was calculated as

$$
\hat{d}_{j} = \frac{1}{N_{R}} \sum_{i=1}^{N_{R}} \left\| {}^{R}P_{ij} - \frac{1}{N_{R}} \sum_{i=1}^{N_{R}} {}^{R}P_{ij} \right\|_{2}, \tag{7}
$$

where $N_{R}$ corresponds to the number of sampled poses of the robotic arm at the same sampling distance.

Referring to the official parameters of the D435i camera and considering the actual picking situation, experiments were grouped according to sampling distance and divided into $x$ groups. For each group, the distance sampled was $z$ = 400, 500, 600, 700, 800, 900, and 1000 mm. For each group, the "world coordinates" control mode of the robot arm was used and the *z*-axis values were limited while the *x,y* coordinates and Euler angles *uvw* continuously changed such that the camera took pictures of the checkerboard from 15 different poses. For each successfully acquired corner, the error of each corner was calculated and the time spent each time was recorded. Taking the data at $z$ = 400 mm as an example, its positioning error is visualized in Figure 10. The vertical and abscissa coordinates in the diagram indicate the number of corner points in the checkerboard rows and columns. Respectively, the *z*-axis value was the error of this corner point.

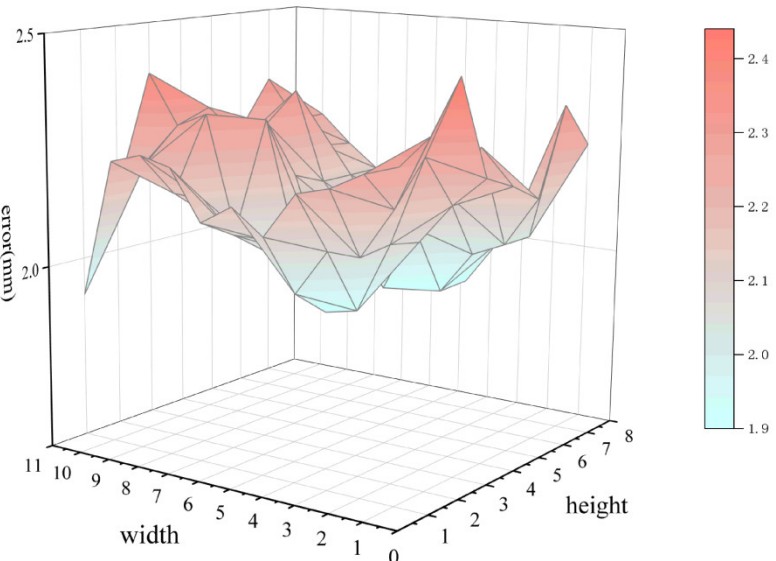

**Figure 10.** Visualization of positioning errors at $z$ = 400 mm.

The results for each capturing distance are shown in Table 1.

**Table 1.** Error indicators corresponding to our method.

| Capturing Distance (mm) | 400 | 500 | 600 | 700 | 800 | 900 | 1000 |
|---|---|---|---|---|---|---|---|
| Maximum error (mm) | 2.44 | 2.78 | 3.92 | 3.82 | 4.48 | 5.07 | 5.95 |
| Standard Deviation (mm) | 0.13 | 0.16 | 0.27 | 0.32 | 0.36 | 0.42 | 0.55 |
| Mean error (mm) | 2.15 | 2.43 | 2.86 | 2.98 | 3.50 | 4.26 | 4.48 |
| Average time per point (ms) | 33 | 32 | 33 | 33 | 34 | 33 | 34 |

A gradual increase in error for the hand–eye system occurred as the sampling distance increased, which was in line with the trend in error for structured-light ranging as the distance changes (Table 1). The error of the whole hand–eye system was seen to be within 6 mm in the range of $z$ < 1 m. There was an end-effector tolerance in the actual situation of fruit picking, while the subsequent long-close distance localization also facilitated the impact of less error. In the meantime, the average time for acquiring a single depth point was 33 ms, which allowed the task to be completed in a short time, even when there was a large number of target objects in the scene. Therefore, the error and time consumption of the entire hand–eye system were within acceptable limits and met the requirements of the picking task.

*3.2. Long-Distance Localization and Picking Experiments*

3.2.1. Experiment for Evaluating the Performance of the Long-Distance Object Detection Network

The uncertainty of fruit size and detection distance during actual vision recognition was simulated by considering the workspace of the robotic arm with sampling distances of 0.8, 1, 1.2, and 1.4 m considered. Multi-angle sampling was carried out to consider possible scenarios, such as lighting and low color contrast. A total of 800 samples were collected and 1500 samples were obtained by data augmentation, which were divided into a training set, a validation set and a test set in the ratio of 8:1:1.

After training, the network model was implemented in Open Source Computer Vision Library (OpenCV) and called under the Windows platform. The predicted bounding

box was obtained by forward propagation, and the spatial coordinates of the centroids were calculated and input to the PCL point cloud space as a separate point cloud. Thereafter, the precision–recall (P-R) curve corresponding to the single class was drawn (Figure 11).

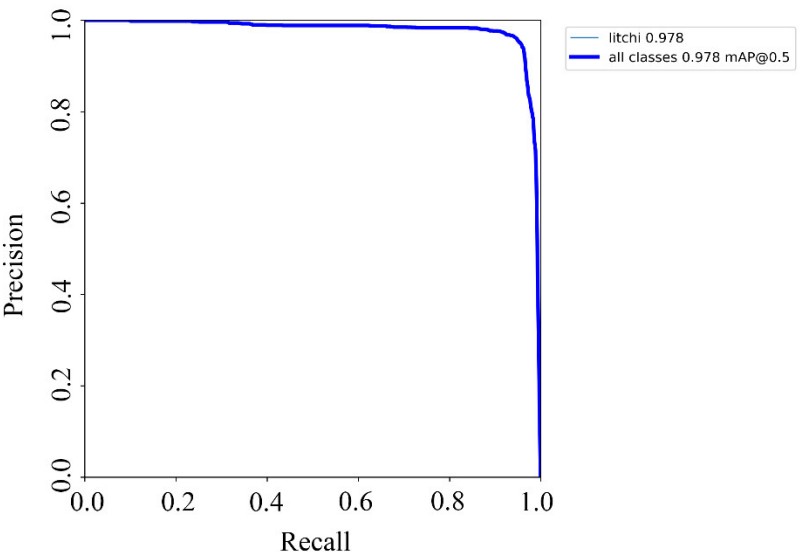

**Figure 11.** P-R curve of the model.

To validate the performance of the model, the average precision (AP) was calculated, expressed as

$$AP = \int_0^1 p(r)dr, \tag{8}$$

where $p$ stands for the precision, $r$ for recall, and $p(r)$ a function of $r$, that is equal to taking the area under the curve (Table 2). The AP of the network was 0.924, indicating that the trained YOLOv5 network had the capability of adapting to multi-angle and complex lighting conditions for litchi recognition.

**Table 2.** Indicators corresponding to our network.

| Method | AP | Recall | Precision |
|--------|------|--------|-----------|
| YOLOv5 | 0.924 | 0.9345 | 0.9159 |

3.2.2. Spatial Point Cloud Density Clustering Performance Experiments

There are various methods for spatial point cloud clustering: division-based clustering, and hierarchical clustering in dividing class clusters based on distance in the clustering process such that they can only be used for spherical clusters. However, realistic litchi bunches do not have a fixed shape. The DBSCAN density clustering algorithm was used here to divide the high-density regions in samples into clusters while filtering out the effects of noisy samples to achieve better results.

The performance of clustering for delineating fruit bunches and the adaptability for proximity scenes were verified through experiments conducted on litchi bunches using Euclidean and DBSCAN density clustering, while finding the optimal setting parameters. Euclidean clustering operates based on Euclidean distance, which is equivalent to the distance between two points in Euclidean space, by arbitrarily choosing a point $p$ in the space, finding the $k$ nearest points to point $p$ through the KD-Tree nearest neighbor search algorithm, and clustering the points whose distance is less than a set threshold into the set $Q$. If the number of elements in $Q$ is no longer increasing, the whole clustering

process is finished. Otherwise, points other than point $p$ are selected in the set $Q$ and the above process is repeated until the number of elements in $Q$ is not increasing.

The experimental objects included 10 bunches of litchi bunches with different shapes and different numbers of litchis. They were marked from 1 to 10 and combined in pairs (Figure 12). The distance between litchi bunches was continuously adjusted and clustered in real time during the experiment and the minimum distance between different clusters at each moment was output. A trained YOLOv5 network was used to detect each litchi individual and calculate the spatial location. The results of the neural network and clustering algorithm were simultaneously output on the screen in real time and visualized (Figure 13).

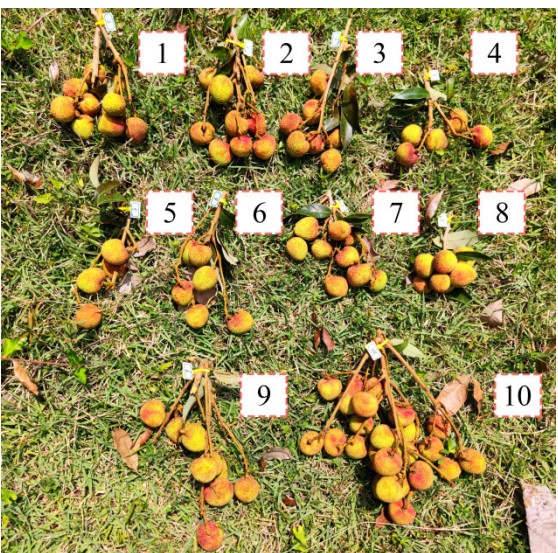

**Figure 12.** Experimental preparation.

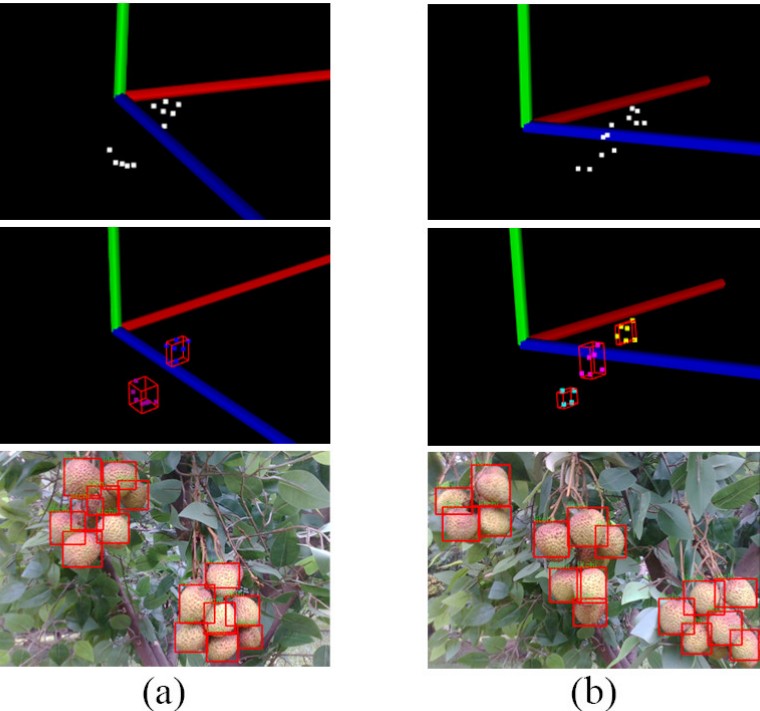

**Figure 13.** DBSCAN experiment: (**a**) two clusters; (**b**) three clusters.

Euclidean and DBSCAN density clustering were then used for clustering. The experiments were approached horizontally and diagonally at 45 degrees, with each pair of bunches approximated five times, and the closest distance between the enclosed boxes for the correct clustering recorded. Experimental results for these two scenarios were visualized (Figure 14).

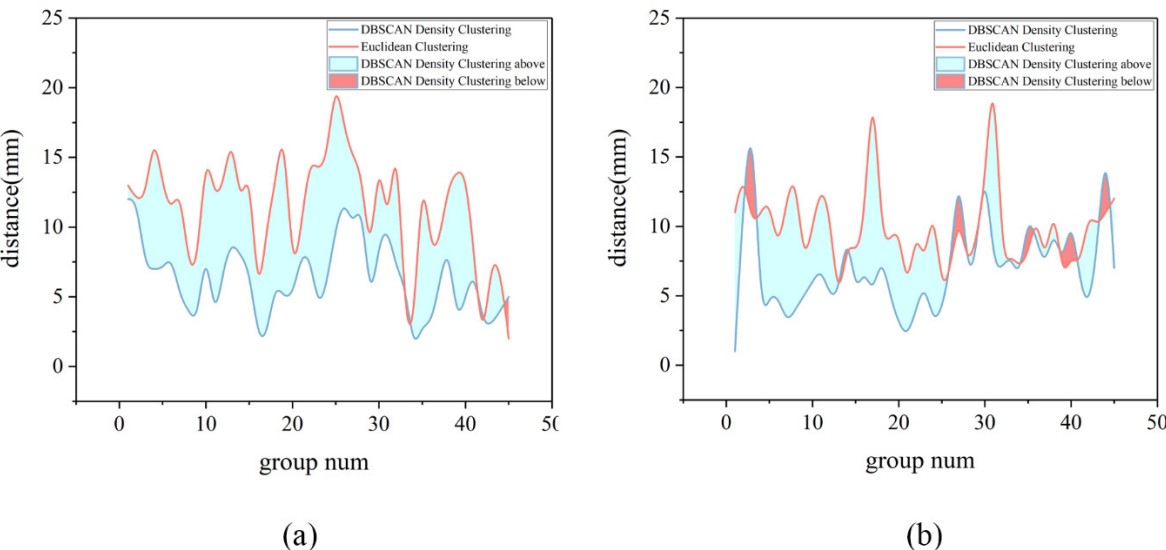

**Figure 14.** Experimental results: (**a**) horizontal; (**b**) diagonal at 45 degrees.

Graphs of the data showed that the minimum intercluster distances were successfully delineated for each fruit combination when using the different clustering methods, while using different colors to indicate the distance relationship between the different methods. In most cases, DBSCAN density clustering was easily seen to be able to successfully divide the different litchi bunches at closer distances, with an average distance of ~7.2 mm and sometimes in contact positions. By looking at a few groups with larger distances, most of them were found to include litchi whose number was 4 or 5. Due to these smaller numbers of litchi fruits and relatively sparse density, the actual results were more easily affected by the denser bunches when using density clustering methods for delineation. Overall, both clustering methods could be used to delineate litchi bunches, while density clustering better reduced interference and was more suitable for delineating litchi bunches with random shapes.

For the vertical distribution, there was a high probability that two bunches of litchis obscured the fruit-bearing branch in a realistic scenario where they were vertical and very close together. Both cluster methods worked well when they were distant from each other such that these conditions were not considered here.

After the clusters were divided, each bunch of litchi was numbered to plan the picking route. The best spatial point for close-distance recognition was found based on the following three conditions. (1) The image contained the fruit-bearing branch. (2) The image contained some individual litchis. (3) The depth distance between the camera and cluster center was within the appropriate operating range of the Realsense D435i.

### 3.3. Close-Distance Fruit-Bearing Stem Location Algorithm Experiments

A robotic arm picking system was set up outdoors, integrating long-close distance full process identification, positioning, and control codes. The robotic arm movements were guided by the close-distance identification point obtained from the long-distance fruit bunch segmentation experiment to complete the fruit-bearing stem segmentation and positioning operation. The experiment involved all the methods described here and was a comprehensive evaluation of the complete harvesting strategy.

### 3.3.1. MaskRCNN Instance Segmentation Network Training and Evaluation

The robotic arm was controlled to reach the close-distance recognition point of fruit bunches by long-distance positioning. A total of 1200 samples were collected, and the Labelme program was used for labeling.

Among these, the labeling method was as follows: for litchi bunches, the requirement was to label the complete fruit-bearing branch as well as the inclusion of litchi fruit, and the area had to be without holes. For bifurcated stems, the requirement was to use quadrilaterals for labeling and the area ratio of the bifurcated part to the unbifurcated part at ~1/2. When the individual in the image was too small, labeling was avoided to prevent more errors in depth acquisition.

After samples were randomly shuffled, they were divided into training, test, and validation sets in the ratio of 8/1/1. After 400,000 iterations of training, the network's loss curve smoothed out and achieved good segmentation performance. The mean average precision (mAP) was calculated using

$$mAP = \frac{\sum_{j=1}^{N_k} \int_0^1 p_j(r)dr}{N_k},\tag{9}$$

where $N_k = 2$ is the number of class and $p_j(r)$ the P-R curve corresponding to the $j$thclass, with the mAP value reaching 0.8 at the condition of 0.5 Intersection over Union.

Real and model litchi strings were trained separately using different datasets. The final loss curves and segmentation results are shown in Figure 15.

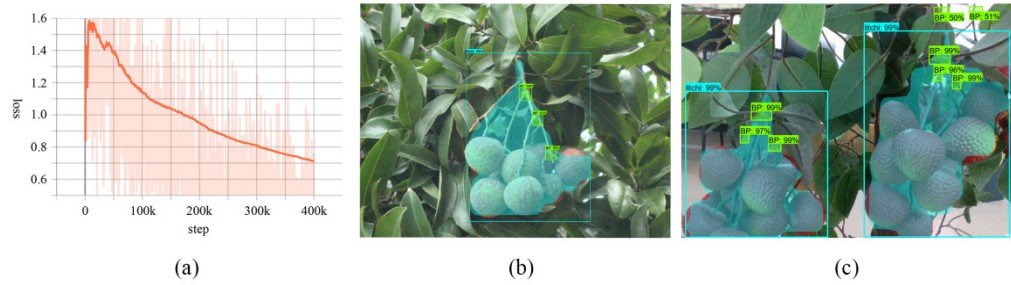

(a)              (b)              (c)

**Figure 15.** Results of MaskRCNN: (**a**) loss curve; (**b**) real scenarios; (**c**) model tree.

### 3.3.2. Equipment and Initialization

A robot platform, equipped with an eye-in-hand structured-light measurement vision system, was built. The platform included an industrial 6-DOF robotic arm, D435i depth camera, end-effector for fruit stem clamping and cutting, and a laptop (Intel Core i7-10750H CPU (Intel Corp., Santa Clara, CA, USA), NVIDIA GeForce RTX 3060 Laptop GPU (NVIDIA Corp., Santa Clara, CA, USA), 16 GB ddr4 RAM (Corsair Corp., California, CA, USA), and Microsoft Windows 10). In addition, a high-precision calibration board and calibration block were used for offline calibration of the robotic arm's hand–eye system.

The robotic arm and control cabinet were fixed to the platform, an end-effector was mounted on the robotic arm flange, and the depth camera was fixed to the end-effector in an upward position to avoid screen occlusion during image acquisition (Figure 16). A laptop computer was used to process the vision data and exchange information with the robotic arm in real time via the transmission control protocol (TCP) port. Prior to an experiment, the end-effector was adjusted to a position horizontal to the ground and the code set to stop and return to the initial position if the system did not detect a fruit bunch or fruit-bearing stem. The camera had a resolution of 1280 × 720 and a sampling rate of 30 fps, set automatically.

In an experiment, the number of individual litchi contained in each string was output through long-distance localization, and then they were picked in order from most to least.

The experiment was judged to be successful when the fruiting branch was finally reached correctly, and the corresponding data were recorded.

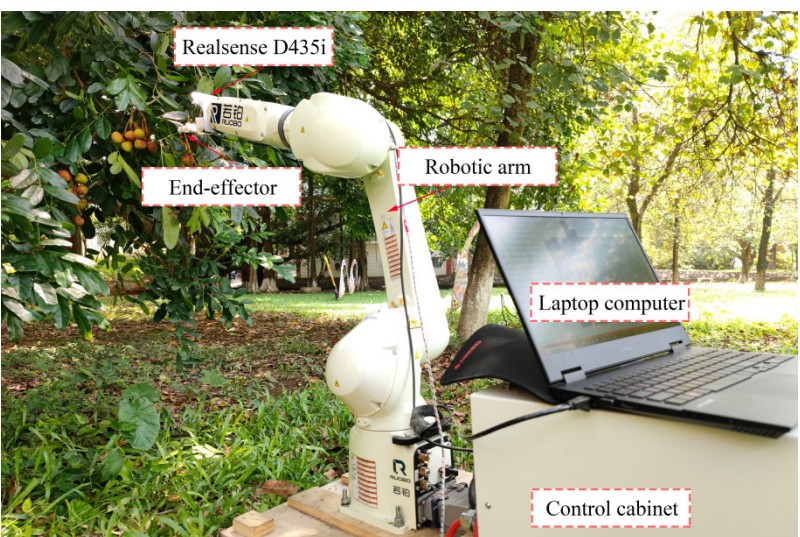

**Figure 16.** Photograph of the platform.

### 3.3.3. Experiments and Results

When conducting experiments on locating and identifying fruit-bearing stems, the steps following steps were followed:

1.  Single sampling at the close-distance identification point and passing it into the trained MaskRCNN model for fruit string and bifurcate stem segmentation;
2.  Extracting the litchi string mask whose center was closest to the center line of the screen;
3.  Traversing all bifurcate stems and if the mask center point was contained within the litchi string mask, then categorizing the bifurcation branches belonging to the litchi string;
4.  Keeping the uppermost bifurcation branch and extracting the part of the litchi string mask above the bifurcation branch for binarization;
5.  Extracting the bifurcation branch mask center point and refining the fruit-bearing stem mask to obtain the positioning line;
6.  Calculating the fruit-bearing stem's pose and reference depth.

The key steps of the process are shown in Figure 17.

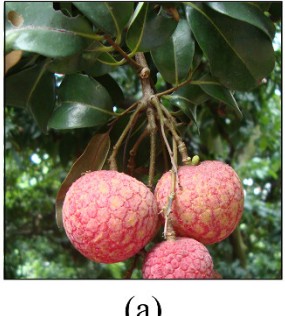 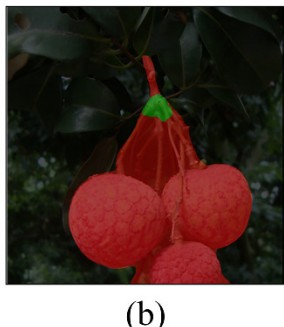 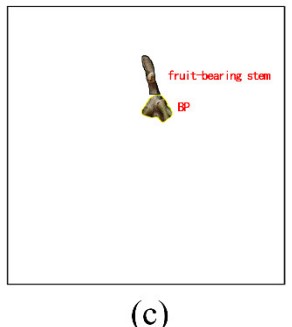 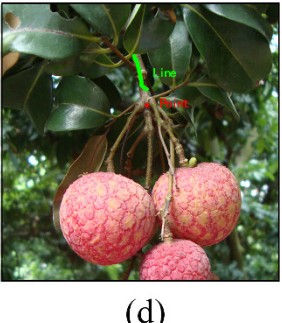

(a) (b) (c) (d)

**Figure 17.** Key steps of the process: (**a**) original image; (**b**) centermost string; (**c**) bifurcation branch and fruit-bearing branch mask; (**d**) final result.

A total of 52 rounds of close-distance identification was performed, with the location of the close-distance picking point set to be the most suitable when the depth camera was 100 mm from the smallest enclosing box of the fruit bunches. Additionally, the height was at the same level as the top surface of the box. At this point, in the field of view, the fruit-bearing stem was located in the upper part of the frame, while ensuring that there were some individual litchis in the frame to better segment the bunches.

Segmentation results in experiments were classified into the following four basic categories: correct positioning, incorrect positioning (offset or not found), intermittent, and positioning line too long or too short; corresponding cases are schematically shown in Figure 18. By examining the processing images of each category, the segmentation results of MaskRCNN were seen to be the main influencing factor, with the quality of the mask of the fruit-bearing stem region directly affecting the effect of refinement. For the two cases of too short and too long, the impact on picking was actually small due to the depth reference point being the main bifurcated stem and the end-effector still reached the picking point within the error range. In the case of intermittence, the overall orientation of the fruit-bearing stem could still be judged, and the longest section was selected as the reference line such that it had less influence on positioning. To avoid collisions between the end-effector and bifurcated stem due to a short positioning line, the top point of the positioning line was chosen as the actual picking point.

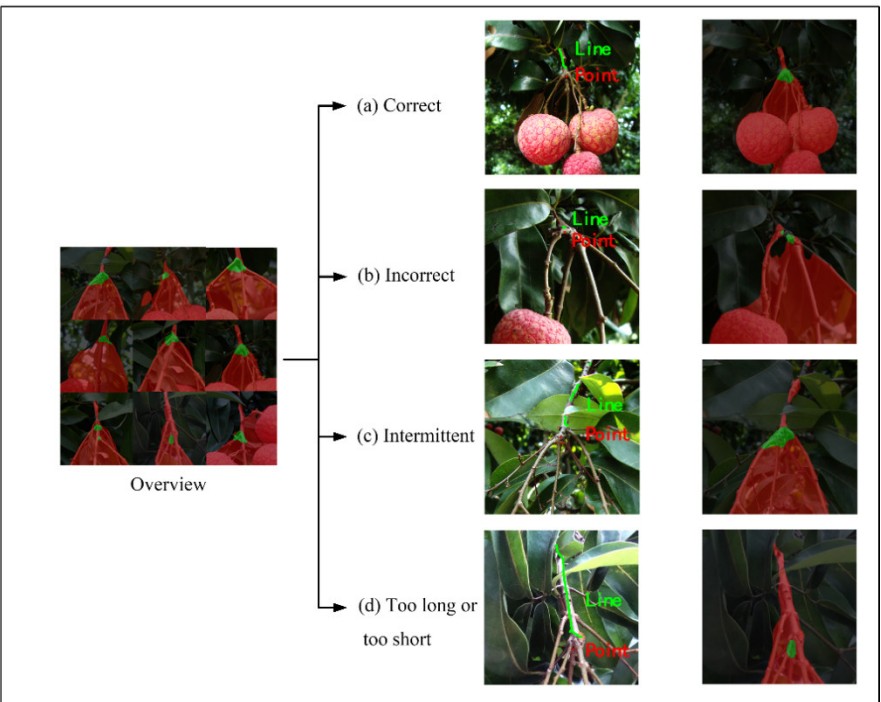

**Figure 18.** Four categories: (**a**) correct; (**b**) incorrect; (**c**) intermittent; (**d**) too long or too short.

The picking results and records of the average time taken to locate the stems showed that all reached the near-field recognition point and the fruit bunches were well segmented (Table 3). The close-distance identification accuracy was found to be in the range of 88.46%, and the average processing time using the CPU was 1.52 s.

**Table 3.** Results of close-distance fruit-bearing stem location algorithm experiments.

| Number of Experiments | Number of Successes | Success Rate | Average Time (Using CPU) |
|---|---|---|---|
| 52 | 46 | 0.8846 | 1.52 s |

After analyzing the failure scenarios, it was found that the recognition effect of bifurcated branches and depth acquisition were the most important factors leading to failure,

especially with the uppermost bifurcated branches. Even if the bifurcation branch in the background was identified, the inclusion relationship between litchi mask and bifurcation mask ensured the stability of the algorithm to a certain extent. The structured-light ranging method was prone to mislocalization when facing small targets, and the depth reference point method was better for guiding the robot arm. The experimental results also showed that the method and time consumption could meet the actual picking requirements in terms of bunch recognition.

### 3.4. Discussions

The methods mentioned above can be useful for other picking robots targeting bunches of fruits. The use of consumer-grade depth camera not only increases the efficiency of detection, but also reduces the economic costs. Clustering method can convert individual fruits into spatial points. This method reduces computational effort and can be applied in other situations, while the depth of picking points is difficult to obtain. All these factors provide theoretical support for the study of other fruit and vegetable picking robots with good economic applications.

Future work will involve algorithm optimization and the production of high-quality datasets. More data sets will be collected, and more segmentation networks will be tested. Or the network structure might be changed to address the problem of poor learning for small bifurcate stems, such that it has a stronger learning capability for fruit features [41–44]. Deep learning with traditional vision algorithms will also be addressed for extracting targets from complex agricultural backgrounds and reducing location error in deep learning [45]. In addition, robotic arms will be installed on mobile devices suitable for the litchi growing environment and corresponding picking experiments conducted in real orchard environments.

### 4. Conclusions

Fruit- and vegetable-picking robots adapted to bunches of fruit objects have been a key research area for researchers in terms of irregular shape interference in bunches of fruit and the localization of fine fruit stalks, as opposed to the type of robot whose target is a single fruit. To improve adaptation to the environment and access to information on fruit diversity, a practical application of appropriate picking strategy and consumer-grade depth cameras in smart agriculture was believed here to be key to the future development of fruit- and vegetable-picking robots.

A robotic arm equipped with an Intel Realsense series depth camera was built, and a long-close distance coordinated control strategy was adopted to take full advantage of the depth camera as well as to avoid the impact of its disadvantages. The advantage was that, by using spatial clustering algorithms, the impact of the difficulty of locating the irregular shape of the bunches was reduced. Even if other types of bunches were replaced, it was still effective and provided positional reference information for the identification of fruit stems. The use of depth reference points and stem positioning lines reduced the positioning errors, and the combined end-effector design tolerances allowed for a better picking operation.

**Author Contributions:** Conceptualization, H.W. and Y.L.; methodology, Y.L.; software, Y.L.; validation, Y.L. and Z.W.; formal analysis, Y.L. and X.X.; investigation, H.W., Y.L., X.X., Z.C. and Y.T.; resources, H.W., Y.L., X.X., Z.C. and Y.T.; data curation, H.W. and Y.L.; writing—original draft preparation, Y.L.; writing—review and editing, H.W., Y.L. and Y.T.; visualization, Y.L.; supervision, H.W. and Y.T.; project administration, H.W. All authors have read and agreed to the published version of the manuscript.

**Funding:** This research was funded by the National Natural Science Foundation of China (Nos. 32071912), Rural Revitalization Strategy Project of Guangdong (20211800400092).

**Data Availability Statement:** The data presented in this study are available on request from the corresponding author. The data are not publicly available due to the privacy policy of the organization.

**Acknowledgments:** The authors would like to thank the anonymous reviewers for their critical comments and suggestions for improving the manuscript.

**Conflicts of Interest:** The authors declare no conflict of interest.

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
