# Peer review of "A Study on Long-Close Distance Coordination Control Strategy for Litchi Picking"

_agronomy, doi:10.3390/agronomy12071520_

Round 1

Reviewer 1 Report

Dear Authors,

In my opinion, the manuscript needs to be corrected, below are detailed comments:

1) Figure 10. Visualization of positioning errors ... - improve the readability of the "error" axis, the maximum should be 2.5, not 3.0 (this will increase the clarity of the graph)

2) Figure 11. P-R curve of the ... - improve legibility, larger font,

3) Figure 14. Experimental…. - as above,

4) You should emphasize and emphasize throughout the work the applicability of the solution and future economic applications,

5) General remark to the result part of the experiment: giving the mean value of the result would be more accurate if you additionally provided the value of the standard deviation (especially since you do not use statistical analysis in your manuscript)

6) Chapter 4 Conclusions and future work - in my opinion it should be entitled "Summary"

7) Pay attention to similar solutions already published (The Concept of the Constructional Solution of the Working Section of a Robot for Harvesting Strawberries, Kurpaska SÅ‚awomir, Bielecki Andrzej, Sobol Zygmunt [et al.], Sensors, 2021, vol. 21, no. 11, pp. 1-17, DOI: 10.3390 / s21113933), include them in your literature review.

Reviewer 2 Report

The article deals with the topic of agricultural robotization from the point of automated fruit(s) picking machinery and utilization of customized NNs in decision making. Authors present an approach that employed robotic manipulator arm (multi-DOF one) in conjunction with YOLO NN derivative for lichi detection, localization and dislocation. Furthermore, it is highly appreciated effort of implementing and utilization of consumer grade equipment and systems into real-work environments and applications. Thus, implementing Intel depth-determination system into real systems is noteworthy attempt.

 Despite the idea and realization there are some key points where article can be improved, such is:

 - due lack of comparison with other approaches the article title has to include a keyword (in some context) 'Study' to designate article dominant approach in idea development rather improvement of existing approach(es).

 - It is unclearly presented how does YOLO (regardless the version) integrates and what exactly serves to? As is stated right now, it suggests that it has important role to litchi bunches (and consequently litchi stem bifurcation points) localization, without details on: how did you train, concatenate and integrate it into your solution? Be more concise, relevant and exact in incorporating YOLO tool into your proposal, especially in Experiment part.

 - Despite of in-advance restriction annunciation regarding number of experimenting litchi bunches, the experimenting setup has to be assembled either from bunches where litchis aren't 'ideal' for analysis. Couple od bunches with occluded, partially or totally, stems and part of bunches, etc. This is needed to prove or disprove your approach real-scenario application suitability.

 - It is highly unclear how does your proposal behaves and what results it produce when whole assembly (robotic arm + camera) gather/acquire image(s) of the litchi tree in highly contrasted and light dynamic environment such is partially shinned tree leaves with direct sun-beam or in environment with rich natural IR irradiation sources? As you know, such scenarios are crude reality in real-life applications, and it can't be neglected!
